# Exploring the Probiotic Potential of Dairy Industrial-Relevant Lactobacilli

**Maria Teresa Rocchetti** [1], **Pasquale Russo** [2], **Giuseppe Spano** [2], **Letizia De Santis** [3], **Ilenia Iarusso** [3], **Nicola De Simone** [2], **Samira Brahimi** [4], **Daniela Fiocco** [1,*] **and Vittorio Capozzi** [5]

1   Department of Clinical and Experimental Medicine, University of Foggia, 71122 Foggia, Italy; mariateresa.rocchetti@unifg.it
2   Department of Agriculture Food Natural Science Engineering (DAFNE), University of Foggia, 71122 Foggia, Italy; pasquale.russo@unifg.it (P.R.); giuseppe.spano@unifg.it (G.S.); nicola.desimone@unifg.it (N.D.S.)
3   Mediterranea Biotecnologie srl, Via Enrico Mattei 85-87, 86039 Termoli, Italy; ldesantis@mediterranea-srl.it (L.D.S.); research@mediterranea-srl.it (I.I.)
4   Laboratory of Applied Microbiology, Department of Biology, Faculty of Natural Sciences and Life, University of Oran 1 Ahmed Ben Bella, Bp1524 El M' Naouer, Oran 31000, Algeria; sam90.brahimi@gmail.com
5   Institute of Sciences of Food Production, National Research Council (CNR) of Italy, c/o CS-DAT, Via Michele Protano, 71122 Foggia, Italy; vittorio.capozzi@ispa.cnr.it
*   Correspondence: daniela.fiocco@unifg.it

**Abstract:** Usually, the search for new candidate probiotics starts from strain isolation, followed by genotypic and phenotypic characterisations. For the best candidates, the final selection criteria, i.e., an efficient biomass production and the survival to stressful conservation processes, may often represent a bottleneck. The aim of this study is to reverse this classic bottom-up approach, thereby evaluating the *in vitro* probiotic properties of microbes that are already commercialized and employed in the dairy sector. The major advantage of reversing the traditional scheme is to deal with strains that are already suitable for the scale-up at the industrial level. In this work, four lactobacilli strains were analysed, belonging to the species of *Lactiplantibacillus plantarum* (strains PLA and PLA2) and *Lacticaseibacillus rhamnosus* (strains PAR4 and RHM). Both *L. plantarum* strains showed the best survival under simulated oro-gastrointestinal stress; PLA and PAR4 had the strongest inhibitory activity against all the tested harmful bacteria, with the latter strain showing also the highest percentage of Caco-2 adhesion; RHM was the best biofilm producer on abiotic surface. Finally, cell-free surnatants from all the strain cultures exhibited anti-inflammatory action on THP-1 macrophages. For all the studied strains, it is possible to claim beneficial functional properties other than the technological ones for which they are already marketed. The possible use of the four strains in a mixture could represent a strategy to diversify and maximize their beneficial potential. Nonetheless, future studies are necessary to validate *in vivo* the observed beneficial properties and to evaluate any effect of the vehicle product on the probiotic aptitude.

**Keywords:** probiotics; lactobacilli; antimicrobial; Caco-2 adhesion; immunomodulation; biofilm; *Lactiplantibacillus plantarum*; *Lacticaseibacillus rhamnosus*

## 1. Introduction

Microbial resources have been receiving relevant interest for their potential to promote innovation and foster sustainability in the agri-food chains [1,2]. The global proposed biotechnological solutions are vast and can find applications in variable production phases, impacting to different extents, from the primary production to the final consumption [3]. Even if many eukaryotic and prokaryotic microorganisms can be exploited to provide social, economic and environmental advances, the importance of lactic acid bacteria (LAB), and lactobacilli, in particular, is well recognised for their symbiotic relationship with humans [4].

Among the most industrially relevant applications of lactobacilli in the food sector, we can include their exploitation in biocontrol and as a source of probiotics [5]. Probiotics have been defined as "live microorganisms, that when administered in adequate amounts, confer a health benefit on the host" [6]. Health-promoting properties have also been ascribed to probiotic dead cells, and their secreted metabolites collectively referred to as postbiotics [7]. In recent years, numerous scientific advances have been achieved in this field, thus increasing the range of both possible benefits given by probiotics (e.g., balancing of the human gut microbiota, improvement of the immune system, reduction of symptoms of lactose intolerance, enhancement of the nutritional value of the matrix) and their potential targets in the population (e.g., elderly, children, subjects exposed to specific nutritional deficiencies, chronic disease patients) [8–10]. This progress contributes to justifying an annual turnover value estimate of about US $46.5 billion in the last years [11].

In food and beverage fermentations, the long-lasting exploitation of LAB also relies on their capacity to produce several metabolites (e.g., organic acids, hydrogen peroxide, bacteriocins) that have antimicrobial activity against undesired foodborne microbes, such as pathogens and spoilers [12,13]. In the last years, the potential in biocontrol of lactobacilli has been increasingly exploited to design protective cultures that can avoid pathogen contaminations and reduce spoilers development, decreasing the use of chemical preservatives in several food chains, particularly in the field of minimally processed foods [14,15]. Indeed, the possible risks associated with ingestion of chemical preservatives represent a relevant issue in consideration of the recent data on exposure to food additive mixtures [16]. The antimicrobial properties of lactobacilli underlie their use both for bio preservation purposes and as probiotics. Indeed, the release of molecules with anti-pathogenic effects provides the basis for the biological control of undesired microbial species in food matrices and, at the same time, contributes to preventing host gut infections, which is one of the most relevant health benefits driven by probiotics [9]. Besides the antimicrobial activity, other properties are desirable for a probiotic, such as the aptitude to survive the hostile environment of the human gastrointestinal tract (GIT), the capacity to colonise the intestinal mucosa, reinforce gut barrier function, stimulate host cells' immune-response, and to synthesise bioactive molecules.

The studies focused on selecting new lactobacilli, candidate strains for the design of probiotic and protective cultures, usually follow a bottom-up approach, with new isolation, genetic characterisation, phenotyping screening and analysis of *in vitro* probiotic features [17,18]. In some cases, the selection was oriented to screen isolates of food origin, looking for both strains with probiotic attitude and strains with biocontrol potential, both for exploitation in the food industry [5,19]. After screening probiotic features, it is necessary to evaluate the ability of the strains to produce biomass efficiently and to survive the stressful conditions that characterise the conservation processes (e.g., freezing, lyophilisation) [20,21]. This is often a bottleneck in the productive scale-up at the industrial level. This work aims to reverse this approach, proposing, for the first time, the characterisation of i) the antimicrobial potential and ii) the *in vitro* probiotic properties of lactobacilli of dairy origin that are already employed in the dairy sector. The advantage is to select strains that are ready for the market and for which it is possible to claim more beneficial properties (i.e., technological and functional) at the same time. Dairy products are generally considered as 'classic' sources of probiotic bacteria [11]. In particular, this study proposes the characterisation of four strains belonging to two species, namely *Lactiplantibacillus plantarum* and *Lacticaseibacillus rhamnosus*, which are among the most representative in the field of probiotic microorganisms [22].

## 2. Materials and Methods

Trypsin-EDTA, RPMI, Dulbecco's modified Eagle's medium (DMEM) were from Gibco (Carlsbad, CA, USA). Fetal bovine serum, L-glutamine, penicillin, streptomycin, phorbol 12-myristate 13-acetate (PMA), 3-(4,5-dimethylthiazol-2-yl)-2,5-diphenyl tetrazolium bromide (MTT), dimethyl sulfoxide (DMSO), lipopolysaccharides (LPS) from *Escherichia coli* O127:B8,

lysozyme, pepsin, bile salts and pancreatin were purchased from Sigma-Aldrich (St. Louis, MO, USA). de Man, Rogosa, and Sharpe (MRS) broth (Biolife Italiana, Milano, Italy); tryptone soya broth (TSB) and Brain Heart Infusion broth (BHI) were purchased from Oxoid (Basingstoke, UK).

## 2.1. Microbial Strains

*Lactiplantibacillus plantarum* PLA and PLA2, *Lacticaseibacillus rhamnosus* PAR4 and RHM, isolated from raw milk (PLA and PLA2), soft cheese (PAR4) and hard cheese (RHM), were used in the current study. The strains are commercialised by Mediterranea Biotecnologie srl (Termoli, Italy). LAB were grown on de Man, Rogosa, and Sharpe (MRS) at 30 °C [23]. *Staphylococcus aureus* UFG142, *Listeria monocytogenes* CECT 4031 and *Escherichia coli* O157:H7 CECT 4267 have been used as indicator pathogenic strains for testing anti-bacterial activity and were grown in tryptone soya broth (TSB) at 37 °C [24]. Cell-free supernatants (CFS) of the tested LAB strains were obtained after centrifugation ($10,000\times g$, 1 min) of cultures at late exponential phase ($2 \times 10^9$ colony-forming units (CFU) mL$^{-1}$, according to previously generated standard growth curves), and subsequent filter-sterilisation (0.20 µm, Syringe Driven Filter Unit, Levanchimica, Italy). The API 50 CHL kit (Biomerieux, La Balme-les-Grottes, France) was used to measure sugar consumption and fast identification of the bacterial strains [25]. The API 50 CHL kit was inoculated with cultured colonies on MRS agar. The results were used to identify bacterial strains using Biomerieux DB [26].

## 2.2. Genotyping Characterization

The genomic DNA of the isolated lactobacilli strains was purified using the QIAamp DNA Kits (QIAGEN). The genomic DNA was used as a template to amplify and sequence 16S rDNA, resulting in species identification [27]. 16S rDNA sequences are available in the supplementary materials (Supplementary Table S1).

## 2.3. In Vitro Survival in Simulated Oro-Gastrointestinal Transit

The oro-gastrointestinal transit (OGT) tolerance assay was performed in a previously designed gastrointestinal system, in which three compartments and five digestion phases (i.e., t1–t5) are simulated [28,29]. LAB were grown until mid-exponential phase ($OD_{600nm} = 1$) and resuspended into sterile saline solution (0.86% NaCl) (stage t0). In the oral phase, the bacterial suspension was supplemented with a gastric electrolytic solution (6.2 g L$^{-1}$ NaCl; 2.2 g L$^{-1}$ KCl; 0.22 g L$^{-1}$ CaCl$_2$; 1.2 g L$^{-1}$ NaHCO$_3$) pH 6.0 and incubated for 5 min at 37 °C with lysozyme 150 mg L$^{-1}$. Then, the gastric environment was simulated by the addition of pepsin (3 g L$^{-1}$) and the pH was progressively reduced to 3.0 (t2) and 2.0 (t3) with HCl 1M. For the small intestinal phase, pH was neutralised to 6.5, and bile salts (3 g L$^{-1}$) and pancreatin (1 g L$^{-1}$) were added (t4). Finally, the large intestine was mimicked by dilution (1:1 *v/v*) with intestinal electrolytic solution (5 g L$^{-1}$ NaCl; 0.6 g L$^{-1}$ KCl; 0.25 g L$^{-1}$ CaCl$_2$) (t5). The viability was determined by spreading serial dilutions from bacterial samples at the different stages on MRS agar and the survival rate was calculated comparing the number of CFU/mL from unstressed and OGT-stressed samples. Survival rate was expressed as LOG (CFUt(1–5)/CFUt0).

## 2.4. Antimicrobial Activity

The antibacterial activity of the lactobacilli strains was evaluated by determining the pathogen growth inhibition with the agar overlay assay, as previously reported [30,31]. Briefly, 5 mL of cultures of lactobacilli at late exponential phase were spotted on MRS agar and grown for 24 h at 30 °C. Then, 10 mL of TSA soft (0.75% agar, *w/v*) were inoculated (1:100, *v/v*) with an overnight cultures of the target strain, and poured over MRS agar plates. After 24 h of incubation at 37 °C, the radius of the inhibition zones was measured and expressed in mm. After 24 h of incubation at 37 °C, the strains were discriminated according to the inhibition halos surrounding the spots, and classified as strains with no

(-), mild (+), or strong (++) inhibition for halos size lesser than 1 mm, between 1–3 mm or greater than 3 mm, respectively.

### 2.5. Biofilm Formation

The ability to produce biofilms was investigated on the plastic surface as previously described [32]. Briefly, LAB strains were inoculated from glycerol stock (1:1000, *v/v*) in MRS and incubated at 30 °C until the exponential phase. Then, cultures were diluted (1:100, *v/v*) in MRS broth, distributed in 96-well microtitre plates and incubated for five days at 30 °C. At this time, wells were washed twice with PBS, stained with 0.05% (*w/v*) crystal violet for 40 min followed by further three washings. After drying, crystal violet was dissolved in 96% ethanol, and absorbance was measured at 590 nm. Assays were performed in triplicate.

### 2.6. Riboflavin Production

Strains from cryopreserved stock were inoculated (1:1000 *v/v*) in the chemical defined riboflavin-free medium described by Russo et al. [33] and incubated at 30 °C for 24 h. The ability to grow after three subsequent inoculums in this medium indicated prototrophy for this vitamin.

### 2.7. Caco-2 Adhesion Assay

Caco-2 cells were grown in DMEM supplemented with 10% (*v/v*) heat-inactivated fetal bovine serum, 2 mM L-glutamine, 50 U mL$^{-1}$ penicillin, and 50 μg mL$^{-1}$ streptomycin, at 37 °C with 5% $CO_2$. Caco-2 cells were seeded in 96-well cell culture plates ($2 \times 10^5$ cells per well) and grown for 2 weeks, changing the medium every 2 days, in order to obtain steady monolayers. Adhesion assays were performed as previously described [29,34–36]. In brief, the growth medium was replaced with absolute DMEM 24 h prior to the adhesion assay. One hundred μL of mid-exponential phase cultures ($OD_{600nm}$ = 0.6–0.8, corresponding to $5 \times 10^8$ CFU mL$^{-1}$) from each bacterial strain were centrifuged, resuspended in DMEM and incubated with Caco-2 cells (0.1 mL per well) for 1 h, at 37 °C, with 5% $CO_2$ (ratio 1000:1, bacteria to Caco-2 cells). After washing wells with PBS, Caco-2 cells and adherent bacteria were detached by addition of trypsin and resuspended in PBS, then serially diluted and plated onto MRS agar to determine the number of cell-attached bacteria, as CFU per mL. In order to calculate the percentages of adhesion, CFU from washed wells, containing only cell-bound bacteria, were compared with those from control unwashed wells containing both unbound and bound bacteria. Adhesion assays were conducted at least in three independent experiments with triplicate determinations.

### 2.8. Immuno-Modulation of THP-1 Macrophages

The immuno-modulation of THP-1 macrophages was carried out using the CFS obtained from stationary phase (16 h) cultures of the tested lactobacilli strains. The CFS concentration to be used for the immuno-modulation test was determined by a cytotoxicity MTT-based test.

### 2.8.1. MTT Assay

Human THP-1 cells were grown in RPMI supplemented with 10% (*v/v*) heat-inactivated fetal bovine serum, 2 mM L-glutamine ad 50U mL$^{-1}$ penicillin, and 50 μg mL$^{-1}$ streptomycin, incubated at 37 °C in humidified atmosphere with 5% $CO_2$. THP-1 used for MTT assay were seeded at a density $5 \times 10^4$ cells/well in 96-well cell culture plates and differentiated into macrophages using 100 ng mL$^{-1}$ phorbol 12-myristate 13-acetate (PMA), as previously described [37]. Macrophages were incubated with 100 μL of different dilutions of CFS (50, 20, 15, 10 and 5%, (*v/v*)) in serum-free medium for each strain and incubated at 37 °C, 5% $CO_2$, for 24 h. After treatment, CFSs were removed, wells were washed with PBS, and macrophages were incubated with 100 μL MTT-containing medium (0.5 mg/mL MTT in serum-free medium) for 4 h, at 37 °C, 5% $CO_2$ [38]. Then, the supernatants were removed from wells, and DMSO (100 μL) was added to dissolve and visualise the formazan

crystal formed in living macrophages. Absorbance at 595 nm was used to calculate the relative cell viability (%) using a microplate reader (FilterMax F5, Molecular Devices, CA, USA). Macrophages non-treated with CFS were used as control, defining 100% viability. The relative cell viability (RCV) was calculated using the following equation: RCV (%) = [(OD$_{595}$ sample − OD$_{595}$ blank)/(OD$_{595}$ control − OD$_{595}$ blank)] × 100.

2.8.2. Effect of LAB CFS on THP-1 Macrophages

Human THP-1 macrophages, obtained as described above, were incubated with lipopolysaccharides (LPS, 200 ng/mL) from *E. coli* O127:B8, with or without pre-incubation with CFS. Precisely, macrophages were seeded on a 24-wells plate ($5 \times 10^5$ cells per well) and incubated whit 5–10% (*v/v*) CFS of each LAB strain for 20 h, at 37 °C, in 5% $CO_2$. Then, 100 ng mL$^{-1}$ LPS were added, and cells were further incubated for 3.5 h. Negative and positive controls were represented by untreated macrophages and macrophages treated with LPS only, without CFS, respectively. After removing supernatants, macrophages total RNA was extracted using TRIzol reagent (Invitrogen, Carlsbad, CA, USA), quantified and checked for integrity (NanoDrop™ V. 3.7.0, Thermo Scientific, Waltham, MA, USA), and subsequently, reverse-transcribed using QuantiTect® Reverse Transcription kit (Qiagen, Valencia, CA USA). The transcriptional analysis of genes for tumour necrosis factor-alpha (TNF-$\alpha$) and interleukin 8 (IL-8) was carried out by quantitative RT-PCR using QuantiFast Sybr® Green PCR kit (Qiagen) in a real-time instrument (ABI 7300, Applied Biosystem, Foster City, CA, USA), by applying the $^{\Delta\Delta Ct}$ method and using the β-actin transcript level as an internal normalizzator, as already described [28].

*2.9. Statistical Analysis*

Variables are expressed as mean ± standard deviation (SD). Significant differences were assessed using one-way analysis of variance (ANOVA) followed by Fishers Least Significant Difference (LSD) test, and Student's *t* test, with $p < 0.05$ as the minimal level of significance. The Statview software package, SAS (v. 5.0) was used for all statistics.

**3. Results**

*3.1. Identification of the Lactobacilli Strains*

*Lactiplantibacillus plantarum* strains PLA and PLA2 were isolated from raw milk, *Lacticaseibacillus rhamnosus* PAR4 from soft cheese and *Lacticaseibacillus rhamnosus* RHM from hard cheese. The investigated selected strains were Gram-positive, catalase-negative and non-motile rods. Metabolic characterisation, based on sugar fermentation patterns, compared with the observations on the API website [26], classified the strains as *Lactiplantibacillus plantarum* (PLA and PLA2), and *Lacticaseibacillus rhamnosus* (PAR4 and RHM) (data not shown). BLAST analysis of the 16S rDNA genes revealed the following high similarities: *Lactiplantibacillus plantarum* ATCC 14917 (100.00% PLA), *Lactiplantibacillus plantarum* ATCC 14917 (99.93% PLA2), *Lacticaseibacillus rhamnosus* JCM1136 (100.00% PAR4) and *Lacticaseibacillus rhamnosus* JCM1136 (100.00% RHM). The growth of lactobacilli in synthetic medium (MRS broth) and skim milk at 30 °C, reached the cell concentration $1.00 \times 10^9$ CFU/mL in 6–8 h, in batch conditions/study. All four strains are already produced at the industrial scale (Supplementary Figure S1) and commercialised for application in the dairy sector, particularly for their potential in bio-controlling spoilage organisms in the dairy manufacturing chain (Supplementary Table S2).

*3.2. Survival in Simulated Oro-Gastrointestinal Stress (OGI)*

The tolerance of the four LAB strains to the OGI environment was investigated in an *in vitro* model that mimics the stress associated with the oral cavity (presence of lysozyme), to the gastric compartment (low pH and hydrolytic enzymes), and to the intestine (neutral pH and activity of pancreatin and bile salts). The survival rate (Figure 1) revealed significant differences between the strains at all the steps of the OGI assay.

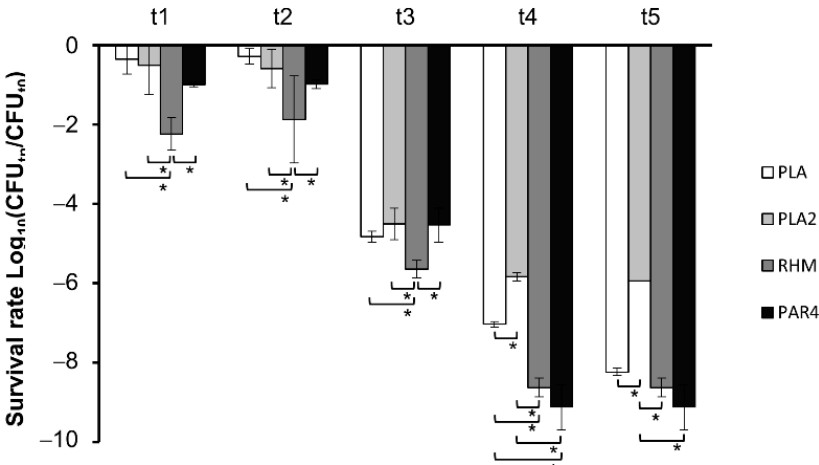

**Figure 1.** Relative survival of the lactobacilli strains at different steps of the *in vitro* simulated OGI transit assay. Data shown are means $\pm$ standard deviations of three different replicates. t1: oral stress (pH 6.0, Lysozyme); t2: gastric stress (pH 3.0, pepsin); t3 gastric stress (pH 2.0, pepsin); t4: Intestinal stress (pH 6.5, bile salts and pancreatin); t5: intestinal stress (pH 6.5). Data shown are means $\pm$ standard deviations. ANOVA test at each time point ($p < 0.05$), followed by Fisher's Least Significant Difference (LSD) test. *: $p < 0.001$.

All the strains showed quite a good tolerance in the first two steps of the oro-gastric conditions (lysozyme, pepsin, and progressive pH downshift from 6.5 to 3.0). At steps t1 and t2, RHM survival decreased by 2.2 and 1.8 log units, respectively, while the other strains retained significantly higher viability at these stages. At step t2, which corresponds to the first exposure to gastric conditions with highly acidic pH, PLA and PLA2 showed higher survival than RHM and PAR4. Upon more acidic condition (t3), RHM resistance was again significantly lower compared to the other strains (i.e., 5.6 log reduction of viability for RHM, vs. 4–5 log reduction for the other lactobacilli). Under intestinal-like conditions (t4 and t5), the differences between survivals tended to reduce, although the two *L. plantarum* strains, and especially PLA2, performed significantly better (i.e., $-5.8$ and $-7.0$ log units at t4, $-5.9$ and $-8.2$ log units at t5, for PLA2 and PLA, respectively) compared to the other LAB strains.

### 3.3. Vitamin B2 Production

The investigated strains were screened for their ability to synthesise vitamin B2 by inoculating three subsequent times in a riboflavin-free medium. Only strain PAR4 was unable to grow in this media, thus suggesting auxotrophy for this vitamin (data not shown).

### 3.4. Antimicrobial Spectrum

The agar spot technique revealed a broad antibacterial spectrum of the strains used in this study. Table 1 reports the data obtained against the foodborne pathogenic bacteria used as indicators. According to the classification proposed by Gaudana et al. [30], the strains analysed exhibited an overall mild-strong antagonistic activity, with inhibition halos ranging from 2.0 to 6.0 radius mm. *E. coli* was the most sensitive target; indeed, the investigated lactobacilli produced broader inhibition halos against this bacterial target. In contrast, the most resistant target was *S. aureus*, which was strongly inhibited only by PLA and PAR4. Strains PLA and PAR4 exhibited a strong antibacterial activity against all the three bacterial targets, whereas *L. plantarum* PLA2 showed a strong antagonistic activity against *E. coli* and *L. monocytogenes*, and, at last, *L. rhamnosus* RHM was antagonistic only against *E. coli*.

**Table 1.** Antibacterial activity against foodborne pathogenic bacteria. Values are expressed as the radius (mm) of the inhibition halos. (+) Inhibition zones between 1–3 mm; (++) inhibition zones greater than 3 mm. Values represent the mean $\pm$ SD of three different experiments.

| Strain | *E. coli* | *S. aureus* | *L. monocytogenes* |
|---|---|---|---|
| *L. plantarum* PLA | ++ (5.5 $\pm$ 1.0) | ++ (3.5 $\pm$ 0.0) | ++ (5.0 $\pm$ 1.0) |
| *L. plantarum* PLA2 | ++ (5.5 $\pm$ 0.5) | + (2.5 $\pm$ 0.5) | ++ (5.0 $\pm$ 1.0) |
| *L. rhamnosus* RHM | ++ (3.5 $\pm$ 0.0) | + (2.0 $\pm$ 0.5) | + (2.0 $\pm$ 0.5) |
| *L. rhamnosus* PAR4 | ++ (6.0 $\pm$ 1.5) | ++ (3.5 $\pm$ 0.5) | ++ (4.5 $\pm$ 0.5) |

### 3.5. Adhesion to Caco-2 Monolayer Cells

Caco-2 monolayer cells were used to test the potential to bind the human intestinal mucosal surface, a valuable probiotic feature. Adhesion abilities differed significantly between the tested lactobacilli. *L. rhamnosus* PAR4 and *L. plantarum* PLA2 showed the highest percentage of adhesion (15.2 $\pm$ 3.5%, 10.2 $\pm$ 1.6%, respectively), while *L. plantarum* PLA and *L. rhamnosus* RHM exhibited a lower ability to adhere to Caco-2 (adhesion rates of 2.2 $\pm$ 0.1% and 2.7 $\pm$ 0.3%, respectively) (Figure 2).

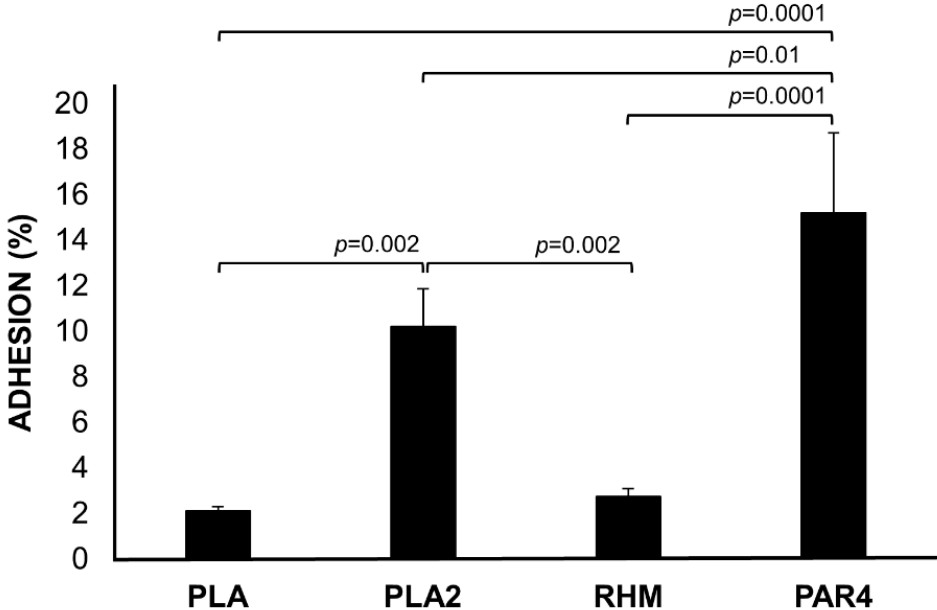

**Figure 2.** Adhesion of the lactobacilli strains to Caco-2 cell monolayers. The adhesion ability of *L. plantarum* PLA and PLA2, *L. rhamnosus* RHM and PAR4 was expressed as the percentage of adhesion. Statistical analyses were carried out by ANOVA test ($p < 0.05$), followed by Fishers Least Significant Difference (LSD) test. Values represent the mean $\pm$ SD of three different experiments.

### 3.6. Biofilm Formation Assays

The ability of the LAB strains to adhere and to form biofilms on plastic surfaces was evaluated. All the strains analysed were able to adhere to the plastic surface as measured by absorbance spectrophotometry (Figure 3). The biofilm formation was strain-dependent, with *L. rhamnosus* RHM as the best biofilm producer on abiotic surface ($OD_{570}$ = 1.69 $\pm$ 0.22) followed by *L. plantarum* PLA ($OD_{570}$ = 0.55 $\pm$ 0.13). On the contrary, *L. plantarum* PLA2 ($OD_{570}$ = 0.05 $\pm$ 0.03) and *L. rhamnosus* PAR4 ($OD_{570}$ = 0.02 $\pm$ 0.01) showed the lowest ability of adhesion.

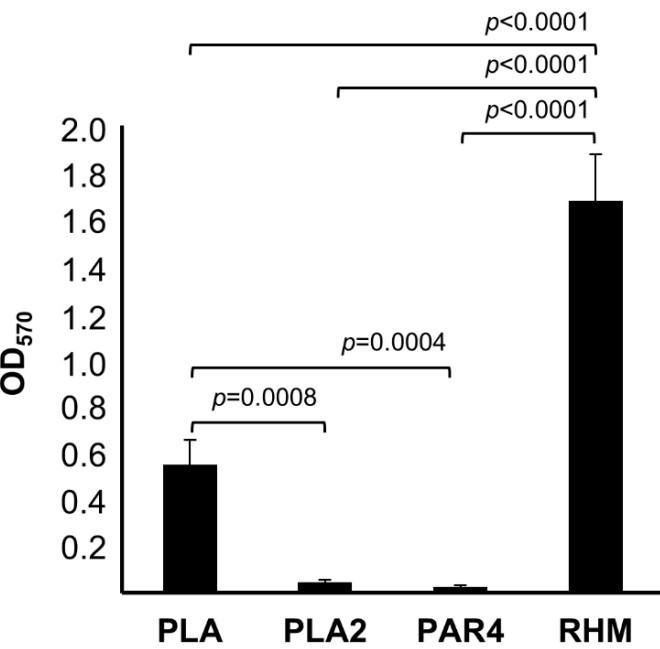

**Figure 3.** Ability to produce biofilms on plastic surface. Cultures of the tested lactobacilli were inoculated in fresh MRS broth and incubated for five days at 30 °C. Optical density was measured at 570 nm to quantify the plastic-adhering biofilm. Statistical analyses were carried out by ANOVA test ($p < 0.05$), followed by Fisher's Least Significant Difference (LSD) test. Values represent mean $\pm$ SD of 3 different experiments.

### 3.7. Immuno-Modulatory Effect of LAB CFS

Cytotoxicity tests, using a range of CSF concentrations to treat macrophages, allowed us to determine safe amounts of CSF, which differed for each strain (Table 2).

**Table 2.** Cytotoxicity studies. Toxic effects of different percentage (15%, 10%, 5% (*v/v*)) of CFS on viability of THP-1 macrophages. Relative cell viability (%) = [(OD$_{595}$ sample − OD$_{595}$ blank)/(OD$_{595}$ control − OD$_{595}$ blank)] × 100. Mean and SD from at least 2 experiments performed in triplicates.

| Relative Cell Viability | 15% CSF | 10% CFS | 5% CFS |
|---|---|---|---|
| *L. plantarum* PLA | 71.0 $\pm$ 5.3 | 91.1 $\pm$ 23.2 | 101.1 $\pm$ 22.4 |
| *L. plantarum* PLA2 | 67.0 $\pm$ 20.1 | 73.9 $\pm$ 28.1 | 82.3 $\pm$ 19.1 |
| *L. rhamnosus* RHM | 55.5 $\pm$ 14.9 | 92.5 $\pm$ 18.5 | 95.3 $\pm$ 23.3 |
| *L. rhamnosus* PAR4 | 109.7 $\pm$ 23.9 | 106.8 $\pm$ 19.6 | 109.0 $\pm$ 10.2 |

The maximal CSF percentages that corresponded to the highest cell viability (cell viability > 80%) were used to test the capacity of the lactobacilli to influence the expression of genes encoding TNF-α and IL-8, i.e., two cytokines with pro-inflammatory and immune-modulatory function. Quantitative RT-PCR was used to evaluate the transcriptional levels of TNF-α and IL-8 genes in LPS-stimulated human macrophages. Pre-treatment of LPS-stimulated macrophages with 5% CFS (*v/v*) from *L. plantarum* PLA and PLA2, and 10% CFS from *L. rhamnosus* RHM and PAR4 CFS led to a decreased relative expression of genes encoding both the pro-inflammatory cytokines, compared to non-treated LPS-stimulated cells (Figure 4). The anti-inflammatory effect of the tested CFS was particularly evidenced by the consistent repression of TNF-α mRNA level, relative to that observed in LPS-stimulated control (Figure 4A).

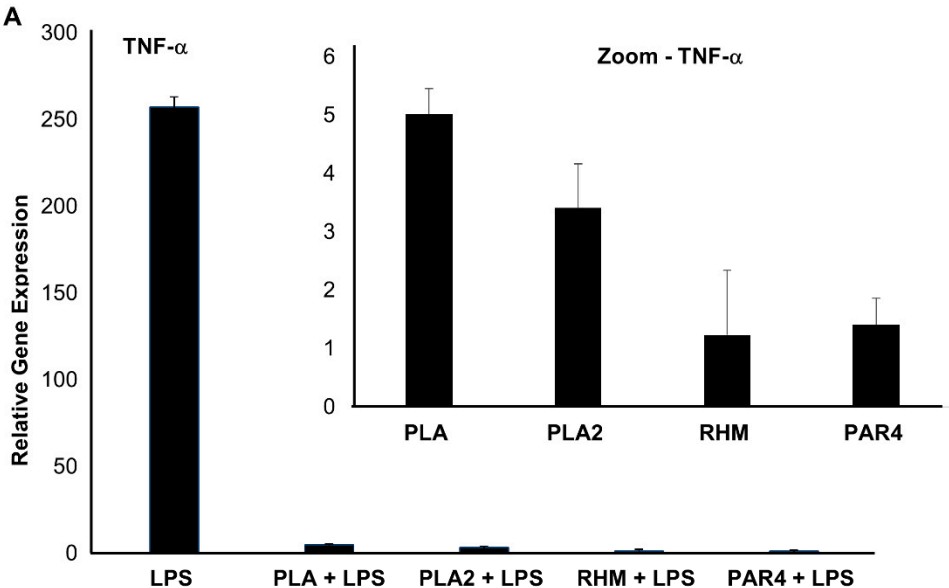

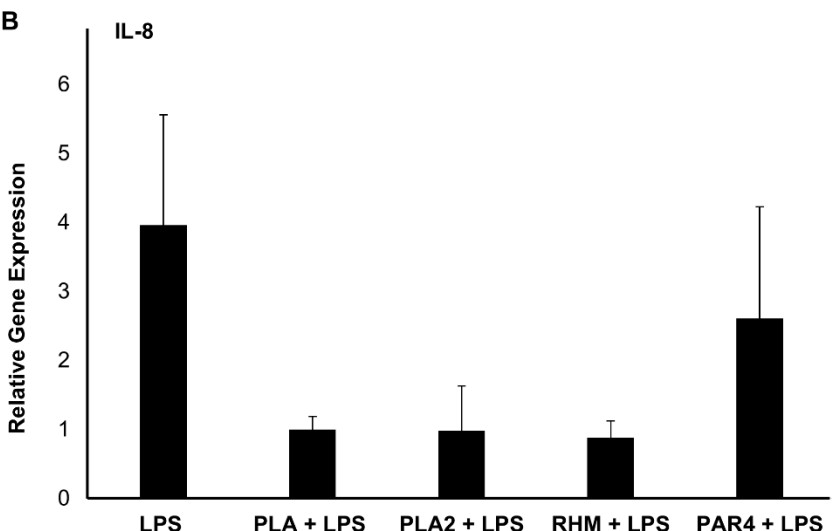

**Figure 4.** Relative transcriptional level of TNF-α (**A**) and IL-8 genes (**B**) in LPS-stimulated macrophages without or with pre-incubation with 5% CFS from *L. plantarum* (PLA or PLA2), 10% CFS from *L. rhamnosus* (RHM or PAR4). Relative gene expression values were obtained by qRT-PCR after 3.5 h of LPS treatment. Relative mRNA level was obtained by normalising to the transcriptional level observed in unstimulated macrophages (gene expression = 1), and β-actin gene was used as an internal control: LPS-stimulated macrophages, LPS (positive control). Values represent the mean ± SD of at least two different experiments. Statistical analyses were carried out by ANOVA test ($p < 0.05$).

Indeed, the transcriptional induction of TNF-α gene was from 51- to 209-fold lower in LPS-stimulated macrophages with CSF-pre-treatment, compared to LPS-stimulated control cells. The same effect was observed for IL-8 gene transcriptional level, which was induced by the pro-inflammatory stimulation with a 2.6- to 5.4-fold reduction in CSF pre-treated macrophages relative to control.

## 4. Discussion

This work aimed to investigate further beneficial properties of lactobacilli that are already commercialised and employed in the dairy sector. Specifically, we investigated the antimicrobial potential and the *in vitro* probiotic properties of four strains belonging to two different LAB species to make them even more attractive and versatile both from

the technological and functional points of view. Ideally, dietary probiotics should fulfil all those actions that can contribute to preserving the balance of the host gut microbiota and prevent dysbiosis [39]. The beneficial effects of probiotics, including lactobacilli, are strain-dependent and not universal; therefore, the investigation of the probiotic criteria is crucial to ascertain health-promoting potential and develop any probiotic product. With this regard, we started to analyse one of the main criteria to select potential probiotic microbes, namely their capability to survive the transit along the gastrointestinal tract, in order to reach and colonise the host intestine [20]. All the tested strains well tolerated the first two steps of the OGT, especially *L. plantarum* PLA and PLA2, whose viability seemed not affected by exposure to oral stress; further, they did not show any significantly decreased viability to the first exposure to gastric conditions, compared to RHM and PAR4. The good survival of *L. plantarum* under oral stress is in agreement with previous studies [40,41]. Even for *L. rhamnosus*, the literature confirms its tolerance to stressors typical of the mouth when studied under similar simulated gastrointestinal condition [41,42]. Persistent exposure to lower pH caused a drop in viability for all lactobacilli, although *L. plantarum* strains, especially PLA2, were much more viable under intestinal conditions, which could be attributed to a higher resistance of this species to acids and bile, compared to the other lactobacilli. A major resistance of *L. plantarum* with respect to *L. rhamnosus* is in accordance with past and recent works that compared probiotic strains' survival upon simulated gastric and intestinal digestion [43,44].

The ability to produce riboflavin is a microbial feature of increasing interest for in situ nutritional status improvements. This phenotype, considered an emerging trait that can be profitably exploited in probiotic strains [45,46], is associated with the capacity to grow in the absence of vitamin B2. A growth under such conditions was registered for all the tested strains, except for *L. rhamnosus* PAR4. This is in accordance with the literature, considering that riboflavin biosynthesis is known to be a strain-specific character within the two investigated species [45,47,48].

Considering the antibacterial potential, *L. plantarum* PLA, along with *L. rhamnosus* PAR4, showed the strongest activity against all the tested harmful bacteria compared to *L. rhamnosus* RHM, which, on the contrary, exhibited strong antagonistic activity only against *E. coli*. The findings confirmed the generally good antimicrobial spectrum observed for *L. plantarum* probiotic strains [49–51]. Notably, PLA and PAR4 strongly inhibited *S. aureus*, i.e., the most resistant among the tested targets, which is known to develop resistance towards conventional antimicrobial agents (i.e., vancomycin and methicillin), causing hazardous health effects worldwide, responsible for increased mortality rate [52,53]. The potential application of *L. plantarum* selected strains for the prevention of staphylococcus-derived infections in humans has already been suggested, albeit mostly by *in vitro* and/or preclinical studies [41,54,55]. In addition to making lactobacilli suitable for clinical purposes, a good antimicrobial activity underlies their uses in the field of food bio-preservation [56]. In fact, especially in the screening of dairy product isolates, both probiotic and biopreservation potentials are explored in a combined form as two applicative aspects of the same biological aptitude [56]. In evaluating desirable probiotic features in the tested strains, we also investigated: (i) their adhesion to human intestinal cells to assess whether they might initiate gut colonization and reinforce the intestinal barrier; (ii) their biofilm-forming capacity, which, when occurring on the intestinal mucosal surface, can promote long-term gut colonisation [57]. Among the tested strains, *L. rhamnosus* PAR4 showed the best adhesion to intestinal cells, as supported by literature data [58,59]. Similarly, *L. plantarum* PLA2 exhibited a good level of adhesion to human enterocytes compared to PLA and RHM, resulting in similar [60] or even higher [40] than other *L. plantarum* probiotic candidates previously assayed. However, a different intraspecific ability of adhesion to human cell lines has been reported in earlier studies [61]. Conversely, compared to the other tested strain, RHM and PLA exhibited the best adhesion capacity on plastic surfaces, hence stressing the diversity between the molecular and cellular mechanisms underlying the microbial adhesion to cells and those involved in the production of biofilms on abiotic surfaces [62,63].

Both epithelial cells and resident macrophages accomplish innate immunity against pathogens during intestinal infection. Indeed, the intestine hosts the largest pool of macrophages, which exert a crucial protective action by phagocytising pathogens and, along with dendritic cells, by secreting interleukins, including the pro-inflammatory TNF-$\alpha$ and IL-8 [64,65]. Here, we investigated whether the production of cytokines by human macrophages could be affected, *in vitro*, by CSF from the tested lactobacilli. The evaluation of the immune-modulatory properties of probiotic-derived substances (rather than viable cultures) complies with the prospect of implementing live cells-free formulas, which can offer some relevant advantages over probiotic products, e.g., greater stability, easier handling and storage in pharmaceutical preparations process, and enhanced suitability for specific targets (e.g., immunocompromised patients) [7]. Several lactobacilli-derived substances have been proved to modulate cytokines expression in different cell types involved in the immune response, representing potential adjuvants in anti-inflammatory therapies [66,67]. All the CFSs analysed in this study seem to possess an anti-inflammatory action, as they drastically contrasted the up-regulation of IL-8 and TNF-$\alpha$ mRNA in LPS-stimulated macrophages. TNF-$\alpha$ is an important cytokine, whose expression in macrophages is triggered in response to pathogens [67]. IL-8 is secreted by several cell types, including macrophages, having chemoattractant and activation properties for inflammatory cells as neutrophils, and it is widely used as an inflammatory marker in intestinal cell cultures [68]. The observed anti-inflammatory effect was quite similar for all tested strains, thus indicating that immune-modulatory properties depend on molecular and cellular features, which are shared among lactobacilli. Our findings are consistent with several *in vitro* studies, which found a decreased TNF-$\alpha$ and IL-8 genes induction upon treating macrophages, or other cells involved in the innate immune response with probiotic lactobacilli CFS or their products (CFS, secreted compound, purified metabolites). For instance, treating LPS-stimulated macrophages with *L. plantarum* CFS resulted in a decreased mRNA level of inflammatory markers, including TNF-$\alpha$ [69]. Interestingly, in earlier works, *L. rhamnosus* CFS was demonstrated to be more effective than live cells in reducing interleukins' secretion by human dendritic cells challenged with *E. coli* [70], thus suggesting that metabolites and compounds secreted by probiotics (e.g., postbiotics) can give a major contribution to their action.

## 5. Conclusions

Overall, for all the strains studied in this work and for their secreted products, it is possible to claim beneficial functional properties other than the technological ones for which they are already marketed. The novelty of our reversed approach for selecting new candidate probiotics, starting from those strains that are already commercialized, could speed up the development of new probiotic formulations. Our findings indicate different aptitudes of the four lactobacilli strains, suggesting that a prospective use in a mixture could represent a strategy to diversify and maximise their beneficial potential. It will be then interesting to investigate this synergistic application towards the design of a multi-strain probiotic. However, future studies will have also to assess the possible effect of the components of the end product on the probiotic properties of the strains. Moreover, it will be necessary to validate the *in vitro* characterization by means of *in vivo* assays.

**Supplementary Materials:** The following supporting information can be downloaded at: https://www.mdpi.com/article/10.3390/app12104989/s1, Figure S1: Industrial scale production of the analyzed strains; Table S1: Genotyping Characterization; Table S2: Antimicrobial activity against food-dairy contaminants.

**Author Contributions:** Conceptualization, M.T.R., P.R., V.C.; methodology, S.B., M.T.R., N.D.S., P.R.; validation, M.T.R., N.D.S., S.B.; formal analysis, M.T.R., P.R.; investigation, M.T.R., P.R., N.D.S.; resources, L.D.S., I.I., G.S., D.F.; writing—original draft preparation, M.T.R., V.C., P.R.; writing—review and editing, M.T.R., V.C., P.R., D.F., G.S.; supervision, G.S., D.F., V.C.; project administration,

G.S., D.F., V.C.; funding acquisition, G.S., D.F. All authors have read and agreed to the published version of the manuscript.

**Funding:** M.T.R. is the beneficiary of researcher's contract for the project no. UNIFG171—CUP D74I19003340002, as part of the initiative "Research for Innovation (REFIN)-POR PUGLIA FESR FSE 2014–2020-Azione 10.4". P.R. is the beneficiary of a grant by MIUR in the framework of 'AIM: Attraction and International Mobility' (PON R&I2014-2020) (practice code D74I18000190001).

**Institutional Review Board Statement:** Not applicable.

**Informed Consent Statement:** Not applicable.

**Data Availability Statement:** We do not have extra data supporting the reported results.

**Acknowledgments:** This work was partially supported by PON project "Conservabilità, qualità e sicurezza dei prodotti ortofrutticoli ad alto contenuto di servizio"—POFACS—CUP B74I20000120005. Thanks is due to Domenico Genchi from the Institute of Sciences of Food Production—CNR for the skilled technical support provided during the realisation of this work.

**Conflicts of Interest:** L.D.S and I.I. are two employees of Mediterranea Biotecnologie srl, the company that commercializes the studied strains in the dairy sector. L.D.S and I.L. had no role in the design of the study; in the collection, analyses, or interpretation of data; in the writing of the manuscript, or in the decision to publish the results.

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
