# Peer review of "Exploring the Probiotic Potential of Dairy Industrial-Relevant Lactobacilli"

_applsci, doi:10.3390/app12104989_

Round 1

Reviewer 1 Report

The authors of the 'potential of diary industrial-relevant lactobacilli' present the benefits of the bacterial strains L. plantaru and L. rham nosus in this manuscript. The experiments are comprehensive and in itself conclusive. Especially, the MIT assays and the oro-gastrointestinal stress results in this manuscript are unarguable an asset for the scientific community. The figures presented are all in coherence with their statistical analysis using ANOVA tests. 

Author Response

We greatly appreciated the comments of the reviewer and we thank him/her for the consideration.

Reviewer 2 Report

The manuscript presented (applsci-1693410) deals with the probiotic potential of dairy industrial-relevant lactobacilli. Overall, the topic is of interest to the scientific community, timely presented, and well organized. The paper is well written and comprehensible. The subject frame of the work is well constructed. So, in this respect and this article should be contributed to present research. I strongly believed that the present manuscript deserves the publication in Applied science after the following revisions.

  1. The manuscript is written in easy understandable language however, there are some typographical mistakes as well. Authors should throughout check the manuscript.
  2. The abstract should begin with a sentence to elucidate the background on which this study was set and then the aim should be mentioned.
  3. The abstract part is too general; I suggest clearly mention limitation, novelty, and your contribution.
  4. This research contains good results and suitable for publication in the journal. But it needs to add some references for the methods.
  5. The results are sufficiently described, substantiated and discussed.
  6. Details about the statistical testing are needed.
  7. I suggest to add SD data in tables
  8. I suggest first time write full name rather than abbreviation; revise throughout in manuscript
  9. Over all, the results are sufficiently described, substantiated and discussed.
  10. I would suggested replace the Figure SEM image with good resolution
  11. I would suggest rewrite the conclusion with clear novelty with separate title.

Author Response

Response: we thank the reviewer for his/her comments.

The manuscript is written in easy understandable language however, there are some typographical mistakes as well. Authors should throughout check the manuscript.

Response: thank you for this indication. In the revised version of MS, we extensively revised the text, correcting many typographical mistakes visible by the “Track Changes” word tool.

The abstract should begin with a sentence to elucidate the background on which this study was set and then the aim should be mentioned. The abstract part is too general; I suggest clearly mention limitation, novelty, and your contribution.

Response: thank you for this comment. In the revised version of MS we changed and integrated the abstract following your suggestions. To avoid confusion, changes in the abstract text are shown in red.

 This research contains good results and suitable for publication in the journal. But it needs to add some references for the methods.

Response: thanks for this suggestion. In the revised MS we added further references to methods.

The results are sufficiently described, substantiated and discussed.

Response: we thank the reviewer for this comment.

Details about the statistical testing are needed. I suggest to add SD data in tables

Response: thanks for this suggestion. In the revised version of MS details about the statistic software have been added, and SD have been added to table 1 (while SD were already present in Table 2)

I suggest first time write full name rather than abbreviation; revise throughout in manuscript

Response: thank you for this comment, the manuscript has been revised according to it.

Over all, the results are sufficiently described, substantiated and discussed.

I would suggested replace the Figure SEM image with good resolution

Response: According to your suggestion we checeked and replaced all the figures, and we also attached them also as separated TIFF files, hoping they appear with a better resolution.

I would suggest rewrite the conclusion with clear novelty with separate title.

Response: Thank you for the suggestion. In the revised version of MS the Conclusions have been rewritten with separate title following your indication and changes in the text are shown in red.

Reviewer 3 Report

Main comment

Four strains - Lactiplantibacillus plantarum PLA and PLA2, Lacticaseiba-27 cillus rhamnosus PAR4, and RHM - isolated from raw milk, soft cheese, and hard cheese were subjected to asses their probiotic potential . The strains were subjected to a survival test in a simulated oro-gastrointestinal system and in vitro assays to test their probiotic effects. The authors claim that all these strains and their secreted products confer beneficial functional properties with different aptitudes. However, the weakness of this study is the lack of in vivo assays, and it is not clear if any end-product containing these strains was also assessed. Food products contain ingredients, additives, and other components that could significantly affect the physiology of probiotic strains. 

Minor comment

Italic fonts in:

Line 99. “Escherichia coli” 

Line 125. “In vitro” 

Line 373. “in vitro

Author Response

Four strains - Lactiplantibacillus plantarum PLA and PLA2, Lacticaseibacillus rhamnosus PAR4, and RHM - isolated from raw milk, soft cheese, and hard cheese were subjected to asses their probiotic potential. The strains were subjected to a survival test in a simulated oro-gastrointestinal system and in vitro assays to test their probiotic effects. The authors claim that all these strains and their secreted products confer beneficial functional properties with different aptitudes. However, the weakness of this study is the lack of in vivo assays, and it is not clear if any end-product containing these strains was also assessed. Food products contain ingredients, additives, and other components that could significantly affect the physiology of probiotic strains.

Response: thank you for your comments. This study was meant to be an introductory in vitro characterisation of industrial strains, but we agree that further studies, including in vivo assays, are necessary to validate their use for probiotic applications. We stress this point in the final lines of the conclusion, among the future perspectives of this work. A brief mention of it is also in the final part of the abstract.

Minor comment

Italic fonts in:

Line 99. “Escherichia coli”

Line 125. “In vitro”

Line 373. “in vitro”

Response: in the revised version of MS all minors have been corrected.

Round 2

Reviewer 3 Report

The manuscript has successfully been improved, clarifying the need for further in vivo studies. Also, other issues were overcome.